# Advance directives prior to COVID-19 diagnosis in a United States national healthcare system

Amy R. Schwartz[1,2]*, Janet P. Tate[2], Lisa Soleymani Lehmann[3,4], Joseph T. King Jr.[5,6], Kristie M. Walenczyk[1,7,8], Woody Levin[1,2], Shelli L. Feder[2,9], Amy C. Justice[1,10,11], Kathleen M. Akgün[1,2]

1 Department of Internal Medicine, Yale School of Medicine, Yale University, New Haven, Connecticut, United States of America, 2 Department of Medicine, VA Connecticut Healthcare System, West Haven, Connecticut, United States of America, 3 Department of Medicine, Harvard Medical School, Boston, Massachusetts, United States of America, 4 Harvard T.H. Chan School of Public Health, Boston, Massachusetts, United States of America, 5 Department of Surgery, VA Connecticut Healthcare System, West Haven, Connecticut, United States of America, 6 Department of Neurosurgery, Yale University School of Medicine, New Haven, Connecticut, United States of America, 7 Department of Psychiatry, Yale University School of Medicine, New Haven, Connecticut, United States of America, 8 Department of Psychiatry, VA Connecticut Healthcare System, West Haven, Connecticut, United States of America, 9 Yale School of Nursing, New Haven, Connecticut, United States of America, 10 Veterans Aging Cohort Study Center, VA Connecticut Healthcare System, West Haven, Connecticut, United States of America, 11 Department of Health Policy and Management, Biomedical Informatics and Data Sciences, Yale School of Medicine, Yale University, New Haven, Connecticut, United States of America

* amy.schwartz2@va.gov

## Abstract

### Background

An advance directive (AD) document allows a patient to indicate their health care preferences and identify an agent to make decisions on their behalf if they lose their ability to communicate. Due to the substantially elevated risk of acute respiratory failure and death during the COVID-19 pandemic, ADs were especially relevant. The objective of this study was to describe AD completed prior to COVID-19 infection (COVID-19) among patients receiving care in a national healthcare system.

### Methods

We conducted a cohort study of United States Veterans Health Administration (VA) patients with COVID-19 between March 2020 and December 2022. AD completion before COVID-19 was ascertained by progress note titles in the electronic health record. Covariates included age, sex, race/ethnicity, marital status, geographic region, health care utilization, calendar quarter of COVID-19, and VA COVID-19 (VACO) 30-day mortality index score.

**Data availability statement:** Due to US Department of Veterans Affairs (VA) regulations and our ethics agreements, the analytic data sets used for this study are not permitted to leave the VA firewall without a Data Use Agreement. This limitation is consistent with other studies based on VA data. However, VA data are made freely available to researchers with an approved VA study protocol. For more information, please visit https://www.virec.research.va.gov or contact the VA Information Resource Center at VIReC@va.gov.

**Funding:** This work was supported by the National Institute on Alcohol Abuse and Alcoholism (P01-AA029545, U01-AA026224, U10-AA013566, U24 AA020794). Support for VA/CMS data provided by the Department of Veterans Affairs, VA Health Services Research and Development Service, VA Information Resource Center (Project Numbers SDR 02-237 and 98-004). The funders had no role in study design, data collection and analysis, decision to publish, or preparation of the manuscript.

**Competing interests:** The authors have declared that no competing interests exist.

## Results

Among 422,028 COVID-19 patients (median age = 62 years; 88.5% male; 58.6% non-Hispanic White (NH-White), 23.3% non-Hispanic Black (NH-Black), 9.2% Hispanic), 67,970 (16.1%) had AD documentation which varied substantially across all covariates. AD completion increased with VACO Index quintiles ranging from 8.2 to 31.0%. In a model adjusted for covariates, relative to NH-White, NH-Black and Hispanic groups had decreased odds for AD (NH-Black odds ratio (OR)=0.77 (95% confidence interval 0.76–0.79); Hispanic OR=0.85 (0.82–0.87)).

VACO index includes age, and both were strongly associated with AD completion. Women compared to men, and those who were widowed, separated/divorced and never married relative to people who were married, had increased AD completion.

## Conclusions

AD completion was overall low, including among patients at high risk of mortality due to COVID-19. When controlling for age, risk for mortality and other covariates, men and people who identify as Black or Hispanic were less likely to have completed an AD. Investment in interventions to facilitate AD completion are needed, particularly among historically underrepresented populations.

## Introduction

Advance directive (AD) completion allows a patient to document their health care preferences and identify a health care agent to make decisions on their behalf if they lose the ability to communicate. It includes the opportunity of stating a specific preference or that they do not have a preference and is ideally accomplished prior to an acute event which may render the patient unable to think clearly or communicate effectively [1]. Completion of an AD has been associated with improved satisfaction with care and decreased surrogate decision maker distress [2–4].

In the United States (U.S.), about one-third of adults complete an AD. Older persons and women have been more likely to complete an AD [1,5–7]. Racial and ethnic disparities in AD completion have been documented in many regional and national samples in the U.S. [8]. Blacks and Hispanics are less likely to complete an AD or have an advanced care planning visit [5,6]. Such disparities are consistently observed when measured by self-report [5,7,9,10] or claims data [6], and in both Medicare [5,6] and general population samples [8,10–12]. While the magnitude of disparities varies across studies, several have estimated AD completion rates nearly twice as high in White compared with Black or Hispanic individuals [5,7–9]; some studies show smaller disparities [6].

The national Veterans Administration (VA) Healthcare System has longstanding policies mandating screening for and encouraging completion of AD. In 2013, VA introduced a standardized progress note title to document AD completion in the electronic health record (EHR) [13]. However, the portion of VA patients who have

completed an AD document is unknown, and disparities in AD completion at VA have not been assessed. The COVID-19 pandemic highlighted the importance of advance care planning especially for those at substantial risk of short-term mortality given infection. However, ADs prior to infection with COVID-19 have not been well described.

We sought to determine what portion of COVID-19 positive (COVID-19) VA patients had evidence of an AD prior to infection. We also asked whether the presence of an AD was predicted by risk of short-term mortality given COVID-19 infection, or by age, sex, race/ethnicity, marital status, geographic region, health care utilization, or calendar time.

## Methods

### Study design and population

This was a retrospective cohort study. Our sample included Veterans with COVID-19 infection, demonstrated by a positive laboratory test [14]. We included EHR data for Veterans who were alive as of Jan. 1, 2020, and engaged in care, defined as having had a primary care visit within 2 years prior to positive COVID-19 test, using the VA Corporate Data Warehouse (n = 7.7 million Veterans). We used the first positive COVID-19 test in inpatient or outpatient settings between March 1, 2020 and December 31, 2022 (n = 446,798). We further restricted analyses to age at diagnosis between 30−100 years and those with the necessary data needed to calculate the VACO Index (n = 422,028) [15,16]. This study was approved by the Institutional Review Boards of VA Connecticut Healthcare System [IRB #1583210−58, 1583212−62] and Yale University [IRB #0309025943], both of whom granted waivers of consent. This cohort study is reported as per the Strengthening the Reporting of Observational Studies in Epidemiology (STROBE) guidelines (S1 File). Authors did not have access to information that could identify participants and data was accessed between 4/21/2021 and 12/12/2024.

### Main measures

Our primary outcome was evidence of AD completion in the EHR between 2014 and date of COVID-19 infection. AD completion was defined as presence of the "Advance Directive" note title in the EHR. In 2013, VHA established this standard progress note title approved to document the entry of an AD document in the patient's record, and the note title indicates that an AD has been completed by the patient and scanned into the EHR. The VA Advance Directive form includes both a Durable Power of Attorney for Healthcare (Health Care Agent) and a Living Will (Preferences about Life-Sustaining Treatments) section; patients may complete one or both. Our definition was restricted to AD completion, rather than a broader definition of AD discussion or screening [17].

Covariates included age at COVID-19, sex, self-reported race/ethnicity (mutually exclusive categories of non-Hispanic (NH) Black, NH White, Hispanic, Asian, American Indian/Alaska Native, Pacific Islander, other, or unknown), marital status (married, separated/divorced, widowed, never married, or unknown), health care utilization (encounters within 2 years of COVID-19 using clinic stop codes [18] [primary care, social work, hospice evaluation/palliative care] and bed section codes for inpatient care), calendar quarter of COVID-19, geographic region (ten mutually exclusive regions as determined by Health and Human Services), and risk of short term mortality given COVID-19 infection measured using the VA COVID-19 (VACO) 30-day index.

The VACO index predicts 30-day all-cause mortality in patients with COVID-19 based on factors present prior to infection including age, sex, and pre-existing comorbidity diagnoses. We chose this index because it is based on information available prior to COVID-19 that might appropriately prompt an AD discussion. It has been validated in a nationwide cohort of U.S. Veterans and in academic medical center and U.S. Medicare cohorts [15,16]. VACO Index quintiles were made to have approximately the same number of AD in each category. Higher quintiles are associated with higher risk of mortality.

Mortality was determined based on data from Social Security Administration, Center for Medicare and Medicaid Services, VHA inpatient deaths and the VA Death Beneficiary database, whose accuracy is comparable to the National Death Index [19,20].

## Statistical analyses

We identified those with COVID-19 and AD completion vs. those without AD using descriptive statistics for measures of central tendency. We compared AD status by patient demographics, marital status, geographic region, healthcare utilization and severity of illness using the VACO Index, using chi-square test for categorical variables and t-test for continuous variables. In exploratory analysis, we evaluated whether associations differed by calendar time. We created 3 categories: Pre-vaccine Mar-Dec 2020, Vaccine Jan-Dec 2021, and Omicron Jan-Dec 2022.

To account for confounding, we examined factors associated with AD completion using multivariable logistic regression modeling guided by subject area knowledge, Akaike information criterion (AIC; lower values indicated better model fit) and area under the curve (AUC and Nagelkerke $R^2$; higher values better model performance). Statistical significance was defined as a two-sided p value < 0.05. All analyses were performed using SAS version 9.4.

## Results

### Patient characteristics and univariable associations with AD completion

Among COVID-19 patients, 67,970 (16.1%, Table 1) had AD identified prior to their positive test result. Median age was 62 years (interquartile range (IQR) 50−73). Most patients were men (88.5%). The majority (58.6%) identified as NH-White, 23.3% identified as NH-Black, 9.2% Hispanic, 5.4% other, and 3.5% unknown; 49.9% were married, 29.5% separated/divorced, 3.7% widowed, and 15.9% never married. In the two years before COVID-19 infection, 70.3% of patients had 10 or more primary care encounters (including vaccinations and telephone care). In the two years before COVID-19 infection, 39.8% had a social work encounter and 2.2% had a hospice evaluation or palliative care encounter; 22.4% had a medical or surgical inpatient admission, and 4.1% had a psychiatric admission.

AD completion increased with each 5-year increment of age (Table 1; Fig 1a). Women were less likely to have AD completed compared with men (13.6% vs. 16.4%, all comparisons statistically significant at <0.001). Patients who identified as NH-White or other were more likely to have AD completion (17.7% and 15.9%, respectively) compared with NH-Black (13.8%), Hispanic (13.1%) or unknown (12.3%). Among categories for marital status, AD was most common among patients widowed (28.7%) compared with 17.7% separated/divorced, 15.3% who were married and 13.2% of those who were never married. AD was more common among those with 10 or more primary care visits (19.5%) as well as those engaged with hospice evaluation/palliative medicine (49.8%) or social work services (25.7%). ADs were also more common among those with any hospitalization during the two years prior to COVID-19, regardless of admission to medical or psychiatric units (31.3% and 29.4%, respectively).

AD completion increased monotonically with risk of mortality given COVID-19 infection based on pre-existing conditions (VACO Index; Fig 1b) and was more common among COVID-19 patients who died within 90 days of COVID-19 (AD among 90-day decedents was 31.8% versus 15.5% among survivors).

The proportion with AD decreased from 18.5% in Mar-June 2020 to 13.6% in Jul-September 2021, then began rising reaching 21.5% in Oct-Dec 2022 (Table 1). This is reflected in the unadjusted odds ratios 1.20 (95% Confidence Interval 1.14–1.27) in Oct-Dec 2022 in (Table 2). This is likely due to older age. In the earliest period 34% were age 70 years or higher, rising to 47% in the latest period. (Table 1).

### Regional variation and AD completion

AD completion was most common for patients receiving care in New England (22.8%; median age = 64) and the Northwest region (18.2%; median age = 62) and least common for Southeast (14.0%; mean age = 61) and mid-Atlantic (13.9%; median age = 62) regions (see Supplementary Table). We further examined by region (VA Integrated Service Network (VISN)) and down to the level of individual facilities; AD completion was as high as 46% and as low as 6% among individual facilities. There was more variation within a given region based on facility-level than between regions (Fig 2).

**Table 1. Patient characteristics (column 2; n = 422,028). Proportion in category with advance directive (column 3; n = 67,970).**

| All n, (%) unless otherwise indicated | All +COVID-19 (column percentage) | Proportion with Advance Directive (row percentage) |
|---|---|---|
| Overall | | 16.1% |
| Age (years), Median (IQR) | 62 (50,73) | 70 (60,76) |
| 30-44 | 78,092 (18.5) | 5,201 (6.7) |
| 45-59 | 106,459 (25.2) | 11,658 (11.0) |
| 60−60 | 94,363 (22.4) | 16,842 (17.8) |
| 70-79 | 108,310 (25.7) | 23,807 (22.0) |
| 80-100 | 34,804 (8.2) | 10,462 (30.1) |
| Sex | | |
| Female | 48,368 (11.5) | 6,591 (13.6) |
| Male | 373,660 (88.5) | 61,379 (16.4) |
| Race | | |
| NH-White | 247,262 (58.6) | 43,842 (17.7) |
| NH-Black | 98,183 (23.3) | 13,558 (13.8) |
| Hispanic | 39,005 (9.2) | 5,129 (13.1) |
| Other | 22,774 (5.4) | 3,619 (15.9) |
| Unknown | 14,804 (3.5) | 1,822 (12.3) |
| Marital Status | | |
| Never Married | 67,189 (15.9) | 8,900 (13.2) |
| Married | 210,729 (49.9) | 32,291 (15.3) |
| Separated/Divorced | 124,326 (29.5) | 21,947 (17.7) |
| Widowed | 15,826 (3.7) | 4,546 (28.7) |
| Unknown | 3,958 (0.9) | 286 (7.2) |
| Quarter | | |
| 2020 Mar-Jun | 11,798 (2.8) | 2,185 (18.5) |
| Jul-Sep | 18,019 (4.3) | 2,568 (14.3) |
| Oct-Dec | 53,375 (12.6) | 8,100 (15.2) |
| 2021 Jan-Mar | 32,137 (7.6) | 4,997 (15.5) |
| Apr-Jun | 9,098 (2.2) | 1,316 (14.5) |
| Jul-Sep | 38,020 (9.0) | 5,171 (13.6) |
| Oct-Dec | 47,024 (11.1) | 6,562 (14.0) |
| 2022 Jan-Mar | 94,232 (22.3) | 14,266 (15.1) |
| Apr-Jun | 35,630 (8.4) | 6,513 (18.3) |
| Jul-Sep | 50,764 (12.0) | 9,450 (18.6) |
| Oct-Dec | 31,931 (7.6) | 6,842 (21.4) |
| Utilization 2 years prior to COVID-19 | | |
| Primary Care | | |
| 1–4 visits | 39,124 (9.3) | 2,515 (6.4) |
| 5–9 visits | 86,332 (20.5) | 7,747 (9.0) |
| 10 or more visits | 296,572 (70.3) | 57,708 (19.5) |
| Social Work | 167.792 (39.8) | 43,047 (25.7) |
| Hospice Evaluation/Palliative | 9,158 (2.2) | 4,557 (49.8) |
| Inpatient Medical | 94,586 (22.4) | 29,577 (31.3) |
| Inpatient Mental Health | 17,309 (4.1) | 5,084 (29.4) |

*(Continued)*

**Table 1.** (Continued)

| All n, (%) unless otherwise indicated | All +COVID-19 (column percentage) | Proportion with Advance Directive (row percentage) |
|---|---|---|
| VACO Index | | |
| <0.05 | 176,913 (41.9) | 14,578 (8.2) |
| 0.05 to <0.10 | 69,729 (16.5) | 10,386 (14.9) |
| 0.10 to <0.15 | 42,577 (10.1) | 7,986 (18.8) |
| 0.15 to <0.20 | 59,063 (14.0) | 12,171 (20.6) |
| ≥0.20 | 73,746 (17.5) | 22,849 (31.0) |
| Mortality | | |
| Survived | 406,252 (96.3) | 62,957 (15.5) |
| ≤30 days | 10,848 (2.6) | 3,303 (30.4) |
| 31–90 days | 4,928 (1.2) | 1,710 (34.7) |

All comparisons of AD status are statistically significant at <0.001.

## Multivariable models

In a fully adjusted multivariable models relative to NH-White, NH-Black and Hispanic groups were associated with decreased odds for AD completion (NH-Black odds ratio (OR)=0.77 [95% confidence interval 0.76–0.79]; Hispanic OR=0.85 [0.82–0.87]). Calendar quarter was also associated with decreased AD completion. Relative to Mar-Jun 2020, each subsequent calendar quarter was associated with decreased AD completion until Oct-Dec 2022 (Table 2, See Model 5). After adjustment for age, race and marital status (Model 3) the increase in AD completion during Oct-Dec 2022 in unadjusted models was attenuated to OR 1.04 (0.98–1.09) and remained so in Model 5 OR 0.96 (0.91–1.02). We found similar associations across the Pre-vaccine, Vaccine, and Omicron eras (data not shown).

Increasing age was associated with AD completion although the effect was attenuated after adjustment for the VACO Index. Healthcare utilization was associated with increased AD completion (5−9 primary care visits during 2 years before COVID vs. 1−4 (reference) OR=1.24 [1.19–1.30]; 10 or more visits OR=2.32 [2.22–2.42]). In contrast to the univariable analyses, female sex was associated with increased odds for AD in adjusted models (OR=1.34 [1.30–1.38]). Relative to people who were married, widowed, separated/divorced, and never married were associated with increased AD completion (Table 2, See Model 5; widowed OR=1.44 [1.38–1.49]; separated/divorced OR=1.29 [1.26–1.31]; never married OR=1.27 [1.24–1.31]).

The VACO Index was strongly associated with AD completion (OR for 2nd quintile of risk 1.73, 95% CI 1.67, 1.79; OR for 5th quintile of risk 5.23, 95% CI 4.98, 5.50). When comparing nested models, from Model 1 which adjusted only for age, race and sex to Model 4 which added marital status, calendar quarter, and VACO index, female sex became a more important predictor of AD. Multivariable models 2–5 showed better fit and discrimination with further adjustment as evidenced by decreasing AIC and increasing AUC and $R^2$ (Table 2). The biggest improvement came from addition of the VACO Index.

## Discussion

ADs allow patients to document their preferences for health care, including identifying a surrogate decision maker before a life-threatening situation occurs, including the opportunity to note that they have no specific preference, thereby supporting the patient's right to autonomy. The COVID-19 pandemic highlights the urgency of identifying ways to make culturally tailored advanced care planning services more accessible. In a national cohort of patients engaged in care, 69% of those at the highest risk of acute mortality from COVID-19 based on their age, sex, and comorbidities, i.e., in the highest VACO

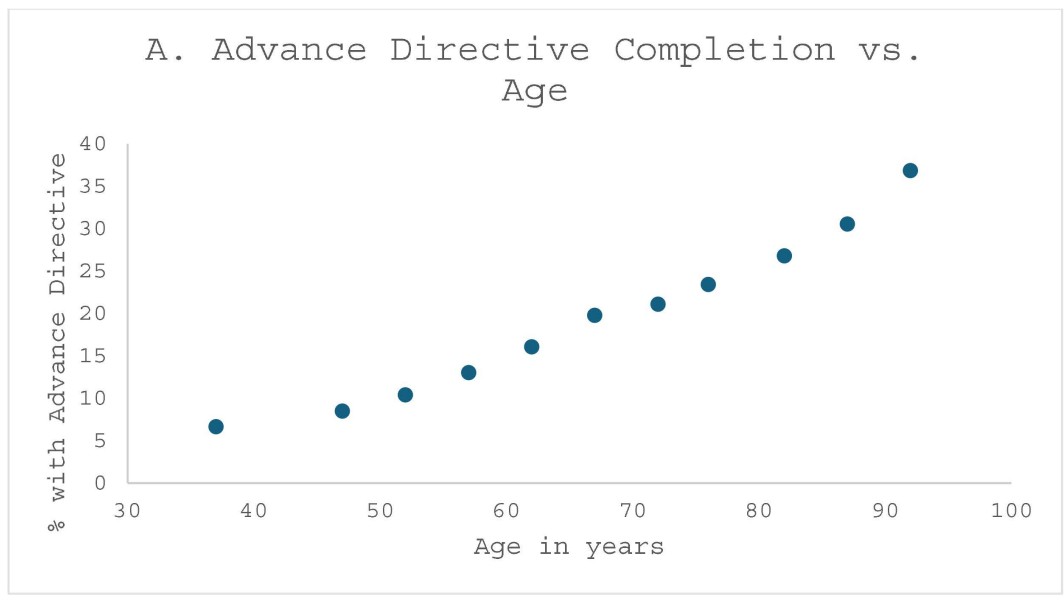

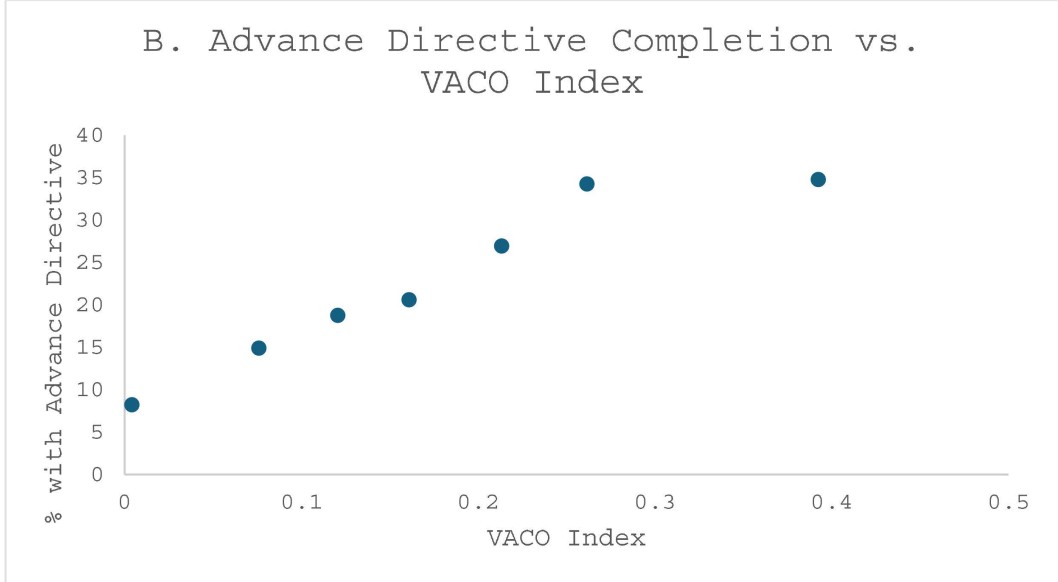

**Fig 1. AD completion by age (Fig 1A) and VACO Index (Fig 1B).**

Index quartile, lacked an AD when they first tested positive. NH-Blacks and Hispanics, who were especially at risk for COVID-19 [21–23], had less AD completion than NH-Whites. Men were substantially less likely than women to have AD completed. In univariable analyses, increasing age was most strongly associated with AD completion. Relative to people who were married, widowed, separated/divorced, and never married were associated with increased AD completion.

There are low rates of AD completion and major disparities in advanced care planning in the U.S [1,8]. The Veterans Health Administration is the largest integrated healthcare system in the U.S. and has a commitment to health equity and respecting patients' preferences for end-of-life care through advance directives [24]; VA has been successful at reducing racial/ethnic health care disparities compared to health care delivered outside the VA [25]. Even in VA, we found low rates

**Table 2. Unadjusted and multivariable associations with AD completion prior to COVID-19, March 1, 2020 – December 31, 2022; subsequent models include previous covariates, with additional covariates included in higher numbered models.**

| | | Unadjusted Base Model | Model 1 Adjusted for age, race, sex | Model 2 Add marital status | Model 3 Add calendar quarter | Model 4 Add VACO index | Model 5 Add primary care utilization |
|---|---|---|---|---|---|---|---|
| AIC | | | 356,312 | 355,218 | 354,675 | 348,218 | 344,538 |
| AUC | | | 0.649 | 0.655 | 0.658 | 0.684 | 0.697 |
| R-Square | | | 0.038 | 0.040 | 0.042 | 0.056 | 0.064 |
| | | OR (95%CI) | OR (95%CI) | OR (95%CI) | OR (95%CI) | OR (95%CI) | OR (95%CI) |
| Age | 30-44 | 1.00 | 1.00 | 1.00 | 1.00 | 1.00 | 1.00 |
| | 45-59 | 1.72 (1.67,1.78) | 1.75 (1.69,1.81) | 1.77 (1.71,1.83) | 1.75 (1.69,1.81) | 1.52 (1.46,1.57) | 1.41 (1.36,1.47) |
| | 60-69 | 3.05 (2.95,3.15) | 3.14 (3.04,3.24) | 3.13 (3.03,3.24) | 3.06 (2.96,3.16) | 1.39 (1.32,1.46) | 1.34 (1.28,1.41) |
| | 70-79 | 3.95 (3.83,4.08) | 4.03 (3.90,4.16) | 4.12 (3.98,4.26) | 3.97 (3.84,4.11) | 0.95 (0.90,1.01) | 0.98 (0.93,1.04) |
| | 80-100 | 6.02 (5.81,6.25) | 6.17 (5.94,6.40) | 6.13 (5.90,6.37) | 5.85 (5.62,6.08) | 0.97 (0.92,1.03) | 1.06 (1.00,1.13) |
| Race | White, NH | 1.00 | 1.00 | 1.00 | 1.00 | 1.00 | 1.00 |
| | Black, NH | 0.74 (0.73,0.76) | 0.84 (0.82,0.85) | 0.81 (0.80,0.83) | 0.80 (0.79,0.82) | 0.78 (0.76,0.80) | 0.77 (0.76,0.79) |
| | Hispanic | 0.70 (0.68,0.72) | 0.87 (0.84,0.89) | 0.87 (0.84,0.90) | 0.85 (0.83,0.88) | 0.86 (0.84,0.89) | 0.85 (0.82,0.87) |
| | Other | 0.88 (0.85,0.91) | 0.99 (0.95,1.03) | 0.98 (0.95,1.02) | 0.97 (0.94,1.01) | 0.96 (0.92,0.99) | 0.94 (0.91,0.98) |
| | Unknown | 0.65 (0.62,0.69) | 0.73 (0.69,0.77) | 0.75 (0.71,0.79) | 0.74 (0.71,0.78) | 0.77 (0.73,0.81) | 0.81 (0.77,0.85) |
| Sex | Male | 1.00 | 1.00 | 1.00 | 1.00 | 1.00 | 1.00 |
| | Female | 0.80 (0.78,0.83) | 1.26 (1.22,1.30) | 1.22 (1.18,1.25) | 1.21 (1.17,1.24) | 1.49 (1.45,1.54) | 1.34 (1.30,1.38) |
| Marital Status | Never | 0.84 (0.82,0.87) | | 1.25 (1.22,1.28) | 1.23 (1.20,1.27) | 1.25 (1.21,1.28) | 1.27 (1.24,1.31) |
| | Married | 1.00 | | 1.00 (0.00,0.00) | 1.00 (0.00,0.00) | 1.00 | 1.00 |
| | Sep/Div | 1.18 (1.16,1.21) | | 1.30 (1.27,1.32) | 1.29 (1.27,1.32) | 1.28 (1.25,1.31) | 1.29 (1.26,1.31) |
| | Widow | 2.23 (2.15,2.31) | | 1.49 (1.43,1.55) | 1.49 (1.44,1.55) | 1.42 (1.37,1.47) | 1.44 (1.38,1.49) |
| | Unknown | 0.43 (0.38,0.49) | | 0.64 (0.57,0.72) | 0.64 (0.56,0.72) | 0.67 (0.59,0.76) | 0.72 (0.63,0.81) |
| Quarter | 2020 Mar-Jun | 1.00 | | | 1.00 | 1.00 | 1.00 |
| | Jul-Sep | 0.73 (0.69,0.78) | | | 0.77 (0.73,0.83) | 0.81 (0.76,0.86) | 0.79 (0.74,0.84) |
| | Oct-Dec | 0.79 (0.75,0.83) | | | 0.79 (0.75,0.83) | 0.83 (0.79,0.88) | 0.80 (0.76,0.85) |
| | 2021 Jan-Mar | 0.81 (0.77,0.86) | | | 0.81 (0.76,0.85) | 0.84 (0.79,0.89) | 0.80 (0.76,0.85) |
| | Apr-Jun | 0.75 (0.69,0.80) | | | 0.79 (0.73,0.86) | 0.83 (0.77,0.90) | 0.79 (0.73,0.85) |
| | Jul-Sep | 0.70 (0.66,0.73) | | | 0.74 (0.70,0.78) | 0.78 (0.74,0.83) | 0.74 (0.69,0.78) |
| | Oct-Dec | 0.72 (0.68,0.75) | | | 0.77 (0.73,0.81) | 0.82 (0.78,0.87) | 0.77 (0.73,0.81) |
| | 2022 Jan-Mar | 0.79 (0.75,0.83) | | | 0.83 (0.79,0.87) | 0.88 (0.84,0.93) | 0.81 (0.76,0.85) |
| | Apr-Jun | 0.99 (0.94,1.04) | | | 0.94 (0.89,0.99) | 1.00 (0.94,1.05) | 0.89 (0.85,0.95) |
| | Jul-Sep | 1.01 (0.96,1.06) | | | 0.93 (0.88,0.98) | 0.97 (0.92,1.02) | 0.87 (0.82,0.92) |
| | Oct-Dec | 1.20 (1.14,1.27) | | | 1.04 (0.98,1.09) | 1.07 (1.01,1.13) | 0.96 (0.91,1.02) |
| VACO Index | <.05 | 1.00 | | | | 1.00 | 1.00 |
| | .05 to <.10 | 1.95 (1.90,2.00) | | | | 1.91 (1.84,1.98) | 1.73 (1.67,1.79) |
| | .10 to <.15 | 2.57 (2.50,2.65) | | | | 2.69 (2.58,2.82) | 2.33 (2.22,2.43) |
| | .15 to <.20 | 2.89 (2.82,2.97) | | | | 3.86 (3.68,4.05) | 3.14 (2.99,3.30) |
| | >=.20 | 5.00 (4.89,5.12) | | | | 6.79 (6.46,7.14) | 5.23 (4.98,5.50) |
| Primary care visits in 2-year interval before COVID | | | | | | | |
| | 1-4 | 1.00 | | | | 1.00 | 1.00 |
| | 5-9 | 1.44 (1.37,1.50) | | | | | 1.24 (1.19,1.30) |
| | 10 or more | 3.52 (3.37,3.67) | | | | | 2.32 (2.22,2.42) |

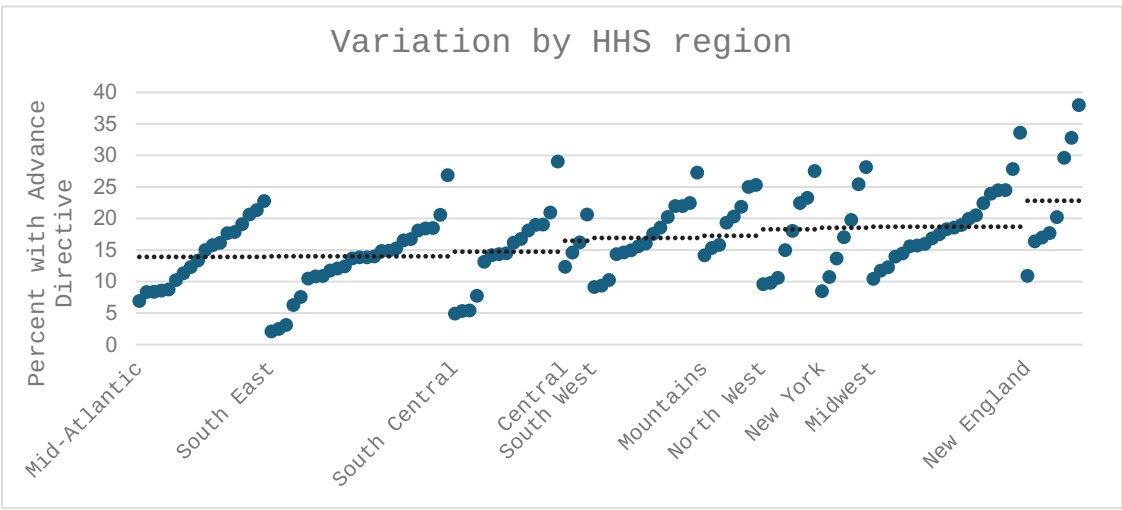

**Fig 2. COVID-19 and AD completion by geographic region and service area within region.** Dashed line = value for each region. Solid circles = VA service areas within region. HHS = Health and Human Services.

of AD completion and racial and ethnic disparities in AD completion. These associations persisted even when adjusting for relevant demographics, marital status (Model 2), time of the pandemic (Model 3) and severity of co-occurring illness as measured by both VACO index (Model 4) and utilization (Model 5). Disparities in AD completion among people who identified as Black or Hispanic suggest a need to adapt AD processes to better meet the needs of racially and ethnically historically underrepresented Veterans.

A 2018 overview of 80 systematic reviews of advanced care planning (ACP) interventions found mostly low-quality evidence for their efficacy. However, they noted that the most successful interventions at increasing AD completion were those that combined computer, video, and discussion elements; those directed to both patients and clinicians; and those providing multiple interactive discussion sessions [26].

A 2021 systematic review identified interventions tested to improve ACP, palliative care, and end-of-life care interventions specifically for racial and ethnic underrepresented groups. They found 13 studies that included an educational intervention focused primarily on advance care planning and/or advance directives [27]. The educational interventions were all designed to improve utilization or access to palliative and end-of-life care for persons from racial and ethnic underrepresented groups by targeting barriers including language, knowledge, attitudes, beliefs, health literacy, and culturally appropriate information delivery. They found that educational interventions targeting underrepresented groups can have positive effects on AD and/or advance care planning related outcomes. However, there were few studies overall, most were quasi-experimental, and only two clinical trials were rated "very high" quality. The existing evidence base for culturally sensitive advanced care planning remains limited. More high-quality intervention studies that address racial and ethnic health disparities in advance care planning are needed. In addition, policy-based approaches to promote ACP could also play a role in AD completion.

Concerns have been raised that AD may not improve goal-concordant care and it is unclear to what degree AD helps patients and surrogates plan for medical decision making [28,29]. However, AD facilitates identification of surrogate decision makers and advanced care planning has been associated with decreased surrogate distress [4,30].

Our study suggests the benefits of using a mortality predictor, such as the VACO Index, to identify patients that should be prioritized for discussions of AD and interventions to improve engagement in ACP. Future directions include understanding the complementary role and implementation of AD and life sustaining treatment (code status) orders. In addition,

it would be important to know if and when COVID-19 patients were prompted to complete AD following infection. While the COVID 19 epidemic is over, we recommend investigation of integrating personalized mortality estimates using multivariable models into prompts for AD discussions.

Our study has several limitations. First, this was a retrospective study on prior AD completion relying on VA documentation of the AD; it may underestimate true AD completion, especially for older Veterans who may be more likely to receive non-VA-based care [31]. It is possible there is residual confounding and unmeasured variables (e.g., socioeconomic status, religiosity) that could have influenced advance directive completion. We also are unable to gauge if the ADs that were completed reflected the preferences and values of the patient for life-sustaining treatments around the time of COVID-19, whether ADs were the product of discussion with a clinician, or if the medical care patients received was concordant with their AD. Finally, while our sample has a large absolute number of women, they represented only 11.5% of the overall sample.

## Conclusions

Among Veterans with COVID-19, advance directive completion was low, including among patients at exceptionally high risk of COVID-19 mortality. People who identify as Black or Hispanic had decreased odds of AD completion. These associations persisted even when adjusting for relevant demographics, marital status, time of the pandemic and risk of mortality given COVID-19 infection. VA may benefit from interventions to facilitate AD completion which integrate personalized risk of mortality estimates with culturally informed communication.

## Supporting information

**S1 Table. +COVID-19 and AD completion by geographic region.**
(DOCX)

**S1 File. STROBE Checklist.**
(DOCX)

## Acknowledgments

This work uses data provided by patients and collected by the VA as part of their care and support. The views and opinions expressed in this manuscript are those of the authors and do not necessarily represent those of the Department of Veterans Affairs or the United States government.

## Author contributions

**Conceptualization:** Amy Schwartz, Janet P. Tate, Lisa Soleymani Lehmann, Joseph T. King, Kristie M. Walenczyk, Shelli L Feder, Amy C. Justice, Kathleen M. Akgün.

**Data curation:** Janet P. Tate, Woody Levin.

**Formal analysis:** Amy Schwartz, Janet P. Tate, Joseph T. King, Amy C. Justice, Kathleen M. Akgün.

**Funding acquisition:** Amy C. Justice.

**Investigation:** Amy Schwartz, Woody Levin, Amy C. Justice, Kathleen M. Akgün.

**Methodology:** Amy C. Justice, Kathleen M. Akgün.

**Writing – original draft:** Amy Schwartz.

**Writing – review & editing:** Amy Schwartz, Janet P. Tate, Lisa Soleymani Lehmann, Joseph T. King, Kristie M. Walenczyk, Shelli L. Feder, Amy C. Justice, Kathleen M. Akgün.

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
