## [Decision Letter · Decision Letter 0]

10 Sep 2025

Dear Dr. Schwartz,

Thank you for submitting your manuscript to PLOS ONE. After careful consideration, we feel that it has merit but does not fully meet PLOS ONE’s publication criteria as it currently stands. Therefore, we invite you to submit a revised version of the manuscript that addresses the points raised during the review process.

The manuscript has been evaluated by two reviewers, and their comments are available below.



We look forward to receiving your revised manuscript.

Kind regards,

Jenna Scaramanga

Staff Editor

PLOS ONE

Journal Requirements:

2. Check the last line of the abstract to ensure it is the same.  The reason for this check is to ensure that the AEs and Reviewers are sent correct information to allow them to make a good decision on whether they can manage/review the manuscript.  Only send back for a change if the abstract on EM and in the manuscript are VASTLY different.

“This work was supported by the National Institute on Alcohol Abuse and Alcoholism (P01-AA029545, U01-AA026224, U10-AA013566, U24 AA020794). Support for VA/CMS data provided by the Department of Veterans Affairs, VA Health Services Research and Development Service, VA Information Resource Center (Project Numbers SDR 02-237 and 98-004).”

Reviewers' comments:

Reviewer's Responses to Questions

**Comments to the Author**

1. Is the manuscript technically sound, and do the data support the conclusions?

Reviewer #1: Yes

Reviewer #2: Yes

2. Has the statistical analysis been performed appropriately and rigorously?

Reviewer #1: Yes

Reviewer #2: Yes

3. Have the authors made all data underlying the findings in their manuscript fully available?

Reviewer #1: Yes

Reviewer #2: Yes

4. Is the manuscript presented in an intelligible fashion and written in standard English?

Reviewer #1: Yes

Reviewer #2: Yes

Reviewer #1: Thank you for allowing me to review this article, which sought to determine what proportion of VA patients who test positive for COVID-19 have evidence of an advance directive prior to infection.

The introduction seems adequate but very brief; I find it lacks a description of AD completion rates and major disparities in advanced care planning in the US. The study design and population are adequately described, as are the measures used. The statistical analyses are presented appropriately, however in the logistic regression models, information on goodness of fit is missing (global Chi², Nagelkerke R², Hosmer-Lemeshow test, etc.). How overfitting was avoided.

In Table 1 of patient characteristics, review the figures and % in some columns (Quarter, Primary Care, VACO Index, Mortality), it would also be good to know the Age (years), Median (IQR) for those who had a completed advance directive. The conclusions supported by the data

Reviewer #2: This study addresses an important and timely topic on advance directive completion among veterans with COVID 19 in the United States. The manuscript is well structured but some revisions would improve clarity, transparency and global readability. The study design should be described consistently as a retrospective cohort study in the abstract, methods and conclusions. The title and abstract should specify the United States so that international readers unfamiliar with the VA system can clearly understand the setting. The abstract wording could be simplified by using phrases such as advance directive completed prior to COVID 19 diagnosis and confidence intervals should be reported in parentheses rather than brackets. The statistical methods require more detail including which tests were used for descriptive comparisons such as chi square or t test and a clear statement that multivariable logistic regression was used with justification for this choice. Descriptive statistics should always specify whether values are means or medians. Results need fuller explanation of the calendar quarter data already shown in Table 1 and the text should clarify whether percentages in tables and figures are row based column based or overall. The flow of the results could be improved by first presenting descriptive findings then unadjusted associations and finally multivariable models. The discussion and limitations section should be expanded to note residual confounding and unmeasured variables which are common in observational studies. Policy implications should be made stronger by discussing how the Veterans Health Administration and the broader United States health system could implement interventions to improve advance directive completion among underrepresented groups. Terminology should also be made consistent by using patients with COVID 19 throughout. Overall this is a valuable and well conducted study and addressing these points will enhance clarity rigor and impact.

**Do you want your identity to be public for this peer review?** For information about this choice, including consent withdrawal, please see our Privacy Policy

Reviewer #1: No

Reviewer #2: **Yes: ** Krupali Patel

---

## [Author Response · Author response to Decision Letter 1]

13 Oct 2025

Please see "Response to Reviewers" document.

---

## [Decision Letter · Decision Letter 1]

27 Nov 2025

Advance directives prior to COVID-19 diagnosis in a United States national healthcare system

PONE-D-25-25607R1

Dear Dr. Schwartz,

We’re pleased to inform you that your manuscript has been judged scientifically suitable for publication and will be formally accepted for publication once it meets all outstanding technical requirements.

Kind regards,

Sk Md Mamunur Rahman Malik

Academic Editor

PLOS ONE

Additional Editor Comments (optional):

Reviewers' comments:

Reviewer's Responses to Questions

**Comments to the Author**

Reviewer #1: All comments have been addressed

2. Is the manuscript technically sound, and do the data support the conclusions?

Reviewer #1: Yes

3. Has the statistical analysis been performed appropriately and rigorously?

Reviewer #1: Yes

4. Have the authors made all data underlying the findings in their manuscript fully available?

Reviewer #1: Yes

5. Is the manuscript presented in an intelligible fashion and written in standard English?

Reviewer #1: Yes

Reviewer #1: (No Response)

**Do you want your identity to be public for this peer review?** For information about this choice, including consent withdrawal, please see our Privacy Policy

Reviewer #1: No

---

## [Editor Report · Acceptance letter]

PONE-D-25-25607R1

PLOS ONE

Dear Dr. Schwartz,

I'm pleased to inform you that your manuscript has been deemed suitable for publication in PLOS ONE. Congratulations! Your manuscript is now being handed over to our production team.

Kind regards,

on behalf of

Dr. Sk Md Mamunur Rahman Malik

Academic Editor

PLOS ONE